

# Joint Measurements of PM$_{2.5}$ and light-absorptive PM in woodsmoke-dominated ambient and plume environments

K. Max Zhang[1], Bo Yang[1], Geng Chen[1,3], Jiajun Gu[1], James Schwab[4], Dirk Felton[5], and George Allen[2]

[1]Sibley School of Mechanical and Aerospace Engineering, Cornell University, Ithaca, NY, USA
[2]Northeast States for Coordinated Air Use Management, Boston, MA, USA
[3]Faculty of Maritime Transportation, Ningbo University, Ningbo, Zhejiang Province, China
[4]Atmospheric Sciences Research Center, University at Albany, State University of New York, Albany, NY, USA
[5]Division of Air Resources, New York State Department of Environmental Conservation, Albany, NY, USA

*Correspondence to:* K. Max Zhang (kz33@cornell.edu)

**Abstract.** DC, also referred to as Delta-C, measures enhanced light absorption of particulate matter (PM) samples at the near-ultraviolet (UV) range relative to the near-infrared range, which has been proposed previously as a woodsmoke marker due to the presence of enhanced UV light absorbing materials from wood combustion. In this paper, we further evaluated the applications and limitations of using DC as both a qualitative and semi-quantitative woodsmoke marker via joint con-

tinuous measurements of PM$_{2.5}$ (by nephelometer pDR-1500) and light-absorptive PM (by 2-wavelength and 7-wavelength Aethalometer®) in three Northeastern U.S. cities/towns including Rutland, VT, Saranac Lake, NY and Ithaca, NY. We compared the pDR-1500 against a FEM PM$_{2.5}$ sampler (BAM 1020), and identified a close agreement between the two instruments in a woodsmoke-dominated ambient environment. The analysis of seasonal and diurnal trends of DC, BC (880 nm) and PM$_{2.5}$ concentrations supports the use of DC as an adequate qualitative marker. The strong linear relationships between

PM$_{2.5}$ and DC in both woodsmoke-dominated ambient and plume environments suggest that DC can reasonably serve as a semi-quantitative woodsmoke marker. We proposed a DC-based indicator for woodsmoke emission, which was then shown to exhibit relatively strong linear relationship with heating demand. While we observed reproducible PM$_{2.5}$-DC relationships in similar woodsmoke-dominated ambient environments, those relationships differ significantly with different environments, and among individual woodsmoke sources. DC correlated much more closely with PM$_{2.5}$ than EcoChem PAS2000-reported

PAH in woodsmoke-dominated ambient environments. Our analysis also indicates the potential for PM$_{2.5}$-DC relationships to be utilized to distinguish different combustion and operating conditions of woodsmoke sources, and that DC-Heating demand relationships could be adopted to estimate woodsmoke emissions. However, future studies are needed to elucidate those relationships.

## 1 Introduction

Woodsmoke resulting from anthropogenic activities is a widespread air pollution problem in many parts of the world. For example, residential woodsmoke is estimated to account for 20% of total stationary and mobile polycyclic organic matter emissions, and 50% of all area source air toxic cancer risks according to the 2011 National Air Toxics Assessment in the U.S.



(https://www.epa.gov/national-air-toxics-assessment). It is reported that around 35% of total PM$_{2.5}$ emissions in the United Kingdom came from domestic wood burning in 2015, while road transport only contributed around 13% of the total PM$_{2.5}$ emissions (DEFRA, 2016). In addition to its contribution to regional air quality, residential woodsmoke may cause significant near-source air quality impacts due to relatively low stack heights and low exhaust temperatures. While in some sense wood

burning products may be considered natural substances, the health effects of wood smoke are found to be comparable to those of fossil-fuel combustion sources (Naeher et al., 2007).

Chemicals that are enriched in woodsmoke relative to other sources have been used to quantify woodsmoke impacts on ambient particulate matter (PM). Among them, levoglucosan, a sugar anhydride derived from the pyrolysis of the major wood polymer cellulose, has been used extensively as a molecular marker for woodsmoke because it is emitted at high concentrations

and relatively stable in the atmosphere (Fine et al., 2001; Simoneit et al., 1999). However, detecting levoglucosan in PM samples at present requires detailed chemical analysis, and the related information is not widely available.

The widely deployed Aethalometer® has made possible continuous aerosol light absorption measurements, commonly referred to as Black Carbon (BC), at either two wavelengths (880 nm and 370 nm) or seven wavelengths (370 nm, 470 nm, 520 nm, 590 nm, 660 nm, 880 nm, and 950 nm). Allen et al. (2004) first proposed using enhanced light absorption of ambient

particulate matter (PM) at 370 nm relative to 880 nm, due to the presence of light absorbing materials from wood combustion near ultraviolet (UV) range, as a marker for woodsmoke PM. Figure 1 depicts the distinct responses of a seven-wavelength Aethalometer (Magee Scientific AE-33) to woodsmoke (Figure 1a) and diesel (Figure 1b) plumes, providing a context for our discussions in this paper. The source of the diesel plume was a backup diesel generator, and the measurement was conducted in 2015. The woodsmoke plume data was collected near a residential woodstove source in early 2016. Note that the purpose of

Figure 1 was to reveal the qualitative differences between the two sources, rather than making a quantitative comparison.

The wavelength-dependent responses to woodsmoke were clearly shown in Figure 1a. At the longer wavelength end, there were virtually no differences in the signals from the 880 nm and 950 nm channels. At the shorter wavelength end, the 370 nm channel recorded the highest reading. We referred to the augmented responses at shorter wavelengths compared to the 880 nm and 950 nm as "UV enhancement". By contrast, virtually no wavelength-dependence (i.e., no UV enhancement) was observed

for diesel exhaust (Figure 1b). There are some slight discrepancies among the different wavelength channels, likely due to the limitations of the real-time dynamic spot loading correction scheme used by the AE-33. The patterns of the wavelength-dependent responses shown in Figure 1 were consistent with the findings from several previous studies, which suggested that UV absorbing compounds are enriched in biomass-combustion PM but scarce in diesel PM (Chen et al., 2015; Olson et al., 2015) or traffic-related PM (Kirchstetter et al., 2004). Broadly, the light-absorbing organic compounds, referred to as "brown

carbon" or BrC, have been shown to strongly absorb UV (Andreae and Gelencsér, 2006).

The concept of DC (also referred to as Delta-C) originated from using the level of the UV enhancement as a marker for woodsmoke PM (Allen et al., 2004). Traditionally, DC was calculated by the differences between 370 nm and 880 nm signals, i.e., DC = BC (370 nm) – BC (880nm), due to the availability of two-channel Aethalometer models. But the concept is not limited to those two particular wavelengths. Figure 1a indicates that woodsmoke UV enhancement started appearing at 660

nm, and more enhancement can be expected at even shorter wavelength (than 370 nm) not available in current Aethalometer





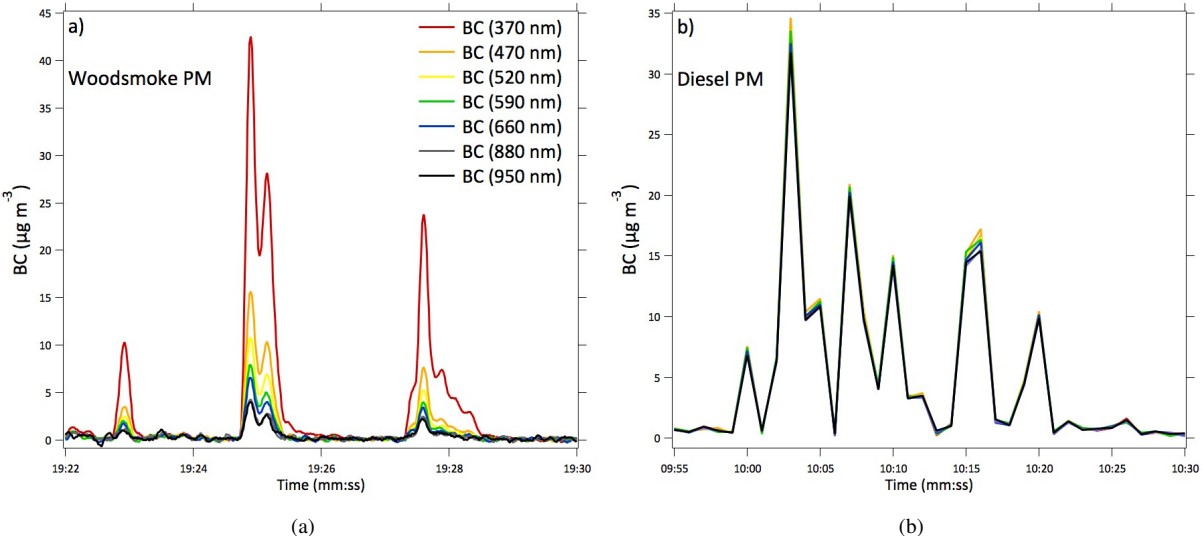

(a)                                                                (b)

**Figure 1.** Wavelength-dependent responses of AE-33 to (a) woodsmoke and (b) diesel plumes. Note that the purpose of this figure is to reveal the qualitative differences, rather than making a quantitative comparison, between the two types of plumes.

models. Studies showed that woodsmoke enhancement peaks at ~300 nm (Kirchstetter et al., 2004; Kirchstetter and Thatcher, 2012). It is possible that including shorter wavelength in future instrumentation development would improve the sensitivity to woodsmoke PM (Olson et al., 2015). Another approach taking advantage of UV enhancement (or wavelength dependence of the aerosol absorption coefficient in general), as reported by Sandradewi et al. (2008a, b), derives light absorption Ångström

exponents ($\alpha$) from multi-wavelength Aethalometer readings, and then use  for quantitative analysis of source contributions to PM. $\alpha$ is close to 1 for traffic sources, and varies for woodsmoke, but generally much larger than 1. This approach often requires light absorption measurements at multiple wavelengths to have a reliable estimate on $\alpha$ (Chen et al., 2015). Since most of the data to be presented in this paper were collected by a two-wavelength Aethalometer, we did not attempt to calculate $\alpha$.

Wang et al. (2011) reported a strong correlation between DC and woodsmoke markers including levoglucosan and elemental

potassium during the heating season, and no statistically significant correlation between DC and vehicle exhaust markers based on field data collected in Rochester, NY. A follow-up study from the same research group used DC as an input variable in source apportionment models, where it was found to play an important role in separating traffic (especially diesel) emissions from wood combustion emissions (Wang et al., 2012). Allen et al. (2011) adopted DC as woodsmoke marker for their fixed-site measurements in Northern New York State, and revealed temporally and spatially resolved patterns of woodsmoke PM (Fuller

et al., 2014). However, Harrison et al. (2013) analyzed data for DC from an Aethalometer network in UK and suggested the presence of other UV absorbing contributors (such as coal burning) to the DC signal. Laboratory experiments conducted by Olson et al. (2015) show that besides biomass burning other sources such as uncontrolled coal (e.g., lignite) and kerosene combustion in lamps can also lead to high DC values. In addition, some secondary organic aerosol (SOA) products have also been found to result in UV enhancement (Zhang et al., 2011; Zhong and Jang, 2011), and increase DC responses.



Motivated by the findings from those previous investigations, we aim to further evaluate the applications and limitations of using DC as a qualitative and semi-quantitative woodsmoke marker. Our work is based on recent joint wintertime measurements of $PM_{2.5}$ and light-absorptive PM in woodsmoke-dominated ambient environments and woodsmoke plume environments in three cities/towns located in the Northeastern U.S. Woodsmoke is known to be the major PM source during wintertime, and

5 predominant PM source during winter nighttime, in the three studied cities/towns. Neither heating by coal nor kerosene lamps are common in this region. Furthermore, SOA formation is typically slow during wintertime. Our study can be regarded as a "necessary condition test" for DC serving as a woodsmoke PM marker. In other words, DC would be deemed an inappropriate marker if it were unable to track woodsmoke PM patterns even under woodsmoke-dominated environments. The paper is organized in such a way that we distinguish the ambient and plume environments by discussing their field measurements and

10 results separately, as the potential implications based on the two types of environments are inherently different. Data from multiple locations and different environments contribute to a more robust evaluation on DC.

## 2   Field Measurements

### 2.1   Woodsmoke-dominated environments: Ambient (Rutland, Clinton and Lakeview) and Plume (Ithaca)

In this paper, we reported the results from field measurements conducted in four sites in three Northeastern U.S. cities, i.e.,

Rutland, VT, Saranac Lake, NY and Ithaca, NY. Table 1 describes the general site characteristics.

**Table 1.** Descriptions of field measurement sites

| Site Name | | Environment | Monitoring Method | Operation Period | Site Descriptions |
|---|---|---|---|---|---|
| Rutland, VT | | Ambient | Fixed-site | October 2011 to June 2013 | Co-located with FEM/FRM at AQS 50-021-0002, no nearby woodsmoke sources |
| Saranac Lake, NY | Clinton | Ambient | | December 2014 to April 2015 | Located in the backyard of a residential property on Clinton Street, minimal woodsmoke sources |
| | Lakeview | Ambient | | January to April 2015 | Located in the backyard of a residential property on Lakeview Street, no nearby woodsmoke sources |
| Ithaca, NY | | Plume | Mobile | December 2015 to March 2016 | Right outside the property lines of woodsmoke sources at downwind direction |





Rutland is the third largest city in the state of Vermont with a population of 16,500, where residential wood combustion is a major source of winter space heating (Frederick and Jaramillo, 2016) and woodsmoke is the dominant PM source in the heating seasons according to the 2014 National Emission Inventory. The ambient air quality monitoring site in Rutland (EPA AQS site number: 50-021-0002) is one of very few routine monitoring stations in the U.S. heavily influenced by woodsmoke

(http://dec.vermont.gov/air-quality/monitoring/network/rutland). Even though Rutland is not a nonattainment area for annual or 24-hr $PM_{2.5}$ National Ambient Air Quality Standards, its $PM_{2.5}$ design value is among the highest in New England. The next two sites were located in Saranac Lake, a rural town of 5,400 people in Upstate New York, where residential wood combustion is the major source of air pollution in winters. Ambient PM concentrations are generally low in Ithaca, the final site and a city of 30,500 in Central New York. While residential wood combustion is not widespread in Ithaca, it has caused

localized air pollution hotspots and complaints against woodsmoke were filed by affected residents living in the densely populated neighborhoods. A primary goal for the field measurements in Ithaca was to capture those hotspots. In short, a common feature among those four sites, given the measurement periods and sampling methods (Table 1), was that they were situated in woodsmoke-dominated environments, and most of the PM could be attributed to woodsmoke sources.

In short, a common feature for the three cities/towns is that woodsmoke is the predominant PM source during winter night-

time, and the only known major source of DC. Furthermore, the Rutland, Clinton and Lakeview sites represent ambient environments since they captured the mixture of multiple sources, not dominated by any one individual source. By contrast, the mobile monitoring technique employed in Ithaca was designed to capture individual sources, thus, representing plume environments.

Table 2 summarizes the major equipment deployed in the different sites. Detailed descriptions of the experimental methods

are provided in Sections 2.2 and 2.3.

**Table 2.** Descriptions of air quality instruments deployed in various field measurements

| Site Names | | $PM_{2.5}$ | Light-absorptive PM | PAH | Others |
|---|---|---|---|---|---|
| Rutland, VT | | pDR-1500 at 5 min time resolution, 2.5 μm cyclone inlet | AE-21 at 5 min time resolution, 2.5 μm cyclone inlet | N/A | FEM and FRM $PM_{2.5}$ monitors |
| Saranac Lake, NY | Clinton | pDR-1500 at 1 min time resolution, 2.5 μm cyclone inlet | AE-42 at 1 min time resolution, 2.5 μm cyclone inlet | EcoChem PAS2000 at 30 s time resolution | 2-D Sonic Anemometer for wind speed and direction |
| | Lakeview | | | | |
| Ithaca, NY | | pDR-1500 at 1 s time resolution, 2.5 μ m cyclone inlet | AE-33 1 s time resolution, 2.5 μm cyclone inlet | N/A | $CO_2$ probe |

The Vermont State Department of Environmental Conservation maintains an air quality monitoring site in Rutland, VT (43.608056° N, 72.982778° W; Elevation: 179 m, EPA site number: 50-021-0002). This site is located in the downtown area





of Rutland, not adjacent to any known woodsmoke sources. Routine measurements of $PM_{2.5}$, $O_3$, CO, $SO_2$, NO, $NO_2$, VOCs and meteorological variables are conducted.

We deployed a personal DataRAM™ Aerosol Monitor (Model pDR-1500, ThermoFisher Scientific, USA) and a two-wavelength Aethalometer™ (370 and 880 nm, Model AE-21, Magee Scientific, USA) for continuous monitoring of $PM_{2.5}$

and Black Carbon (BC), respectively, at the Rutland monitoring site. Operating at 5 min time resolution, both pDR-1500 (1 L min-1, no RH and temperature correction) and AE-21 (2L min-1) were equipped with 2.5 $\mu$m sharp-cut cyclone inlets (BGI model SCC 0.732) placed 1.5 m above the roof of a trailer and ambient air was drawn to the instruments through an aluminum sample line. The pDR-1500 was running from December 2011 to April 2012, during which we were able to compare the $PM_{2.5}$ readings from both pDR-1500 and the collocated Federal Equivalent Method (FEM) instrument (BAM 1020, Met One, USA).

The AE-21 was in operation from October 2011 to June 11, 2013.

All Aethalometer data were corrected for filter spot optical loading saturation effects (Drinovec et al., 2015; Park et al., 2010; Virkkula et al., 2007) using the "binned" approach, first described by Park et al. (2010), as implemented by version 7.1 of the Aethalometer "data masher" program (Allen et al., 2012). This correction provides a more robust measurement of the DC metric, since the optical attenuation for BC at 370 nm is 2.4 times larger than at 880 nm, resulting in a larger loading

artifact at the shorter wavelength. If only BC is present, this results in a negative DC instrument response when the loading is not corrected for.

### 2.1.1 Saranac Lake, NY (Clinton and Lakeview)

Both sites in Saranac Lake, i.e., Clinton and Lakeview, were located in the backyards of residential properties that did not burn wood for either recreational or heating purposes. Both pDR-1500 (1L min-1, no RH and temperature correction) and AE-42

(2 L min-1) were deployed with the same 2.5 $\mu$m sharp-cut cyclone inlets as described in Section 2.2.1, mounted 1.83 m (or 6 feet) above the ground. Both sites were equipped with a 2-D Sonic Anemometer (Model Windsonic, Gill Instruments, UK) for wind speed and direction. In addition, the Lakeview site also included a Photoelectric Aerosol Sensor (Model PAS2000, EcoChem, USA) for continuous particle-bound PAH measurement. The operation periods for the three fixed sites are listed in Table 1.

### 2.2 Mobile Monitoring at Ithaca, NY

As mentioned earlier, we adopted mobile monitoring techniques in Ithaca, NY to identify air pollution hotspots caused by woodsmoke. The sampling inlet of both pDR-1500 and the seven-wavelength Aethalometer (370 nm, 470 nm, 520 nm, 590 nm, 660 nm, 880 nm, and 950 nm; Model AE-33, Magee Scientific, USA), equipped with 2.5 $\mu$m sharp-cut cyclones (BGI SCC 1.197 cyclone at 2.3 L min-1 for pDR-1500; BGI SCC 1.829 cyclone at 5 L min-1 for AE-33), were mounted one foot

above the sunroof of a hybrid electric vehicle (HEV). Although the AE-33 employs automated real-time loading compensation (Drinovec et al., 2015), and thus no post data processing was attempted to account for filter loading effect. To account for the filter loading effect, that correction was not used here, since it is not appropriate for mobile monitoring where different combustion sources are sampled in rapid succession. Filter loading was kept relatively low to minimize any loading effects.





A flow-through type $CO_2$ probe (Model CARBOCAP® GMP343, Vaisala, Finland) was connected to the outlet of AE-33 to record the $CO_2$ level. The pDR-1500 operated without RH correction. RH in the pDR-1500 sensing chamber was always less than 35% without additional sample heating as the instrument was inside a heated vehicle and the chamber temperature was well above ambient dew point. The pDR-1500 was zeroed prior to each mobile run. The pDR-1500 and AE-33 both operated

at 1 s time resolution, and the GMP343 at 2 s time resolution to capture individual woodsmoke plumes.

The mobile monitoring occurred periodically from December 2015 to March 2016. Assisted by the weather forecast from New York State Department of Environmental Conservation (NYSDEC) staff, we chose to conduct mobile runs only during low temperature and low wind speed conditions, when the local air quality impacts from woodsmoke were expected to be significant. We made a total of 20 mobile runs (two in December 2015, seven in January, five in February and six in March

2016). The monitoring routes were recorded at 1 s intervals from a Delorme BU-353S4 GPS receiver using Delorme Street Atlas 2015 PLUS software.

At the beginning of the field campaign, we employed the mobile measurements as an efficient way to survey the air quality levels in the Ithaca area, which then enabled us to identify a few recurring hotspots. The rest of the field campaign focused on those recurring hotspots. Specifically, we parked the HEV right outside the property lines of residential woodsmoke sources in

the downwind direction, and all instruments were powered primarily by the HEV battery without self-pollution. The internal combustion engine of the HEV occasionally turned on to recharge the battery, and caused brief periods of self-pollution. We recorded those conditions, generally characterized by high $CO_2$ and low $PM_{2.5}$ levels, and removed them from subsequent data analysis.

## 3    Results and Discussions

### 3.1    Evaluation of pDR against BAM (and maybe FRM)

As mentioned in Section 2.2.1, we collocated a pDR-1500 with BAM 1020, which is a FEM $PM_{2.5}$ sampler, from December 2011 to April 2012 at the Rutland site. Figure 2 illustrates the comparisons of 24-hour average (Figure 2a), nighttime (10 pm to 6 am) average (Figure 2b), hourly (Figure 2c) and hourly nighttime-only (Figure 2d) $PM_{2.5}$ from the two instruments. The main reason to present the nighttime results was that PM during that period almost exclusively came from woodsmoke sources

in Rutland. Therefore, Figure 1 not only presents the overall comparisons between the two instruments (Figures 2a and 2c), but also how their readings correlated for woodsmoke-dominated environments (Figures 2b and 2d). Note that the apparent horizontal lines in Figure 2c and Figure 2d result from the 1 $\mu g \, m^{-3}$ resolution of the hourly BAM readings.

Table 3 lists the metrics for the regressions. Overall, we found a good agreement between the two instruments. The coefficients of determination, $r^2$, ranged from 0.895 to 0.960. As expected, the daily and nighttime multi-hour averages (0.956 and

0.960, respectively) showed better correlations than hourly and nighttime hourly averages (0.895 to 0.903, respectively). For the hourly data plots, we observed the BAM noise at the origin where pDR-1500 reads very low and the BAM PM is $2 \pm 5 \, \mu g \, m^{-3}$. In general, the comparison results gave us confidence in deploying pDR-1500 for other woodsmoke studies.





**Figure 2.** Comparisons between $PM_{2.5}$ values from BAM 1020 (FEM) and pDR-1500 in terms of (a) 24-hour average, (b) Nighttime (10 pm to 6 am) average, (c) hourly average and (d) nighttime hourly average. The apparent horizontal lines in (c) and (d) result from the 1 $\mu$g m$^{-3}$ resolution of the hourly BAM readings.

The FRM sampler (Model 2025 $PM_{2.5}$ Sequential Air Sampler w/VSCC, R&P, USA) at the Rutland site operates every third day so that we did not include the FRM data in the comparisons. The $PM_{2.5}$ Continuous Monitor Comparability Assessment at the site reported $PM_{2.5}$, FEM = 0.97$PM_{2.5}$, FRM + 1.76 (R=0.97) for Year 2011 and $PM_{2.5}$, FEM = 1.07$PM_{2.5}$, FRM + 0.74 (R=0.92) for Year 2012 (https://www.epa.gov/outdoor-air-quality-data/pm25-continuous-monitor-comparability-assessments).




**Table 3.** Comparisons between BAM 1020 (y) and pDR-1500 (x) from December 2011 to April 2012 in Rutland, VT. The values inside the parentheses represent the corresponding one standard deviation.

|  | Regression | $r^2$ |
|---|---|---|
| **Daily Average** | y = 1.082($\pm$ 0.023)·x+2.12($\pm$ 0.33) | 0.956 |
| **Nighttime average** | y = 1.095($\pm$ 0.022)·x+2.04($\pm$ 0.32) | 0.960 |
| **Hourly average** | y = 1.097($\pm$ 0.007)·x+2.63($\pm$ 0.10) | 0.895 |
| **Nighttime hourly average** | y = 1.040($\pm$ 0.011)·x+2.67($\pm$ 0.16) | 0.903 |

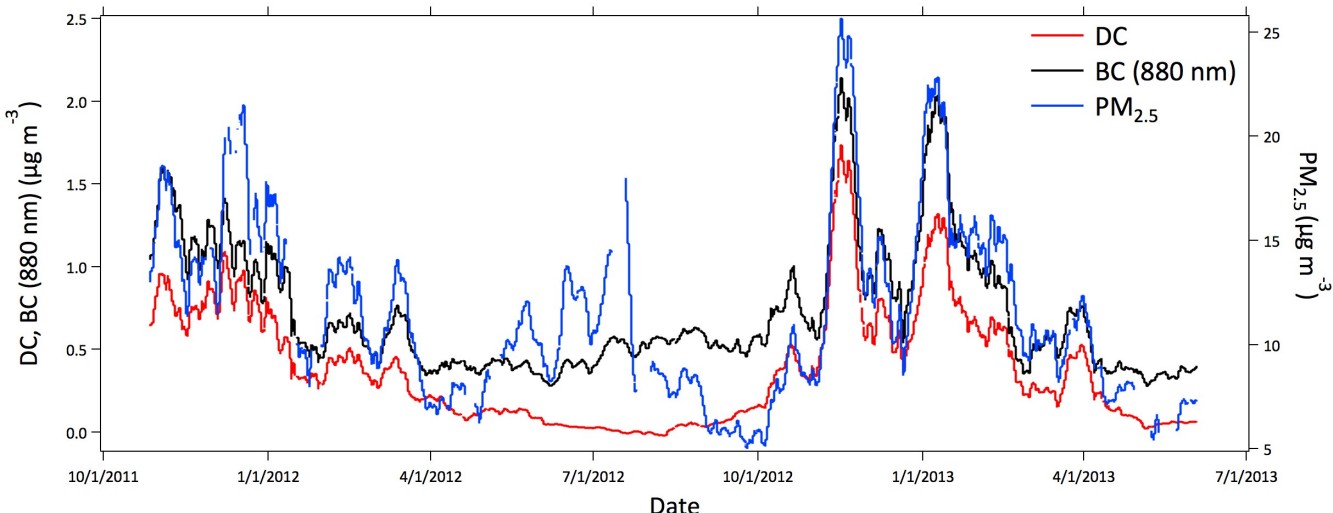

**Figure 3.** Two-week moving average DC (i.e., BC (370 nm)-BC (880 nm)), BC (880 nm), and PM$_{2.5}$ values measured at the Rutland site from October 2011 to June 2013

## 3.2 DC as a qualitative marker for woodsmoke PM

Figure 3 shows the two-week moving average for DC, BC (880 nm), and PM$_{2.5}$ values measured at the Rutland site from October 2011 to June 2013. DC is strongly linked to the season, with highest values in the winter months and much lower values during the summer months. The summertime DC was close to zero, and the non-zero values could be attributed to Canadian forest fires events typically taking place during summer months (Dreessen et al., 2016; Dutkiewicz et al., 2011) and other recreational biomass burning activities. DC, BC (880nm) and PM$_{2.5}$ all peaked at winter months, when they showed very similar temporal trends. This is as expected since a fraction of woodsmoke PM is BC and woodsmoke sources led to high PM$_{2.5}$ concentrations in heating seasons. Nevertheless, unlike DC, the concentrations of BC (880nm) and PM$_{2.5}$ were also significant on occasion in the summertime, likely driven by traffic and other emission sources. This comparison supports DC as a qualitative woodsmoke marker.





Figure 4 illustrates the diurnal variations of DC, BC (880 nm), and PM$_{2.5}$ concentrations, for both summer months (July to September 2012) and winter months (December 2012 to March 2013) at Rutland. As expected, DC showed a strong diurnal pattern in the winter months, elevated during nighttime and peaking around 10 pm, and little variation during the summer months. The diurnal patterns of BC (880 nm) persisted over seasons, but driven by woodsmoke sources in the winter months and

likely by traffic sources in the summer months. The wintertime PM$_{2.5}$ exhibited a strong diurnal pattern, driven by woodsmoke sources, and less significant but still noticeable diurnal pattern in the summertime, driven by traffic sources, which were not as dominant as woodsmoke sources in Rutland, VT. The nighttime enhancement in pollutant concentrations due to changes in the atmospheric boundary layer also contributed to the diurnal patterns both in summertime and wintertime. This comparison further supports DC as a qualitative woodsmoke marker. As mentioned earlier, previous studies found that SOA products may

result in DC signals. However, Figure 4 indicates that SOA is not a main driver for either PM$_{2.5}$ or DC. If SOA formation were significant, we would expect that PM$_{2.5}$ and/or DC would peak around mid-day. The distinct diurnal patterns illustrated in Figure 4 is more consistent with strong influence of local emissions.

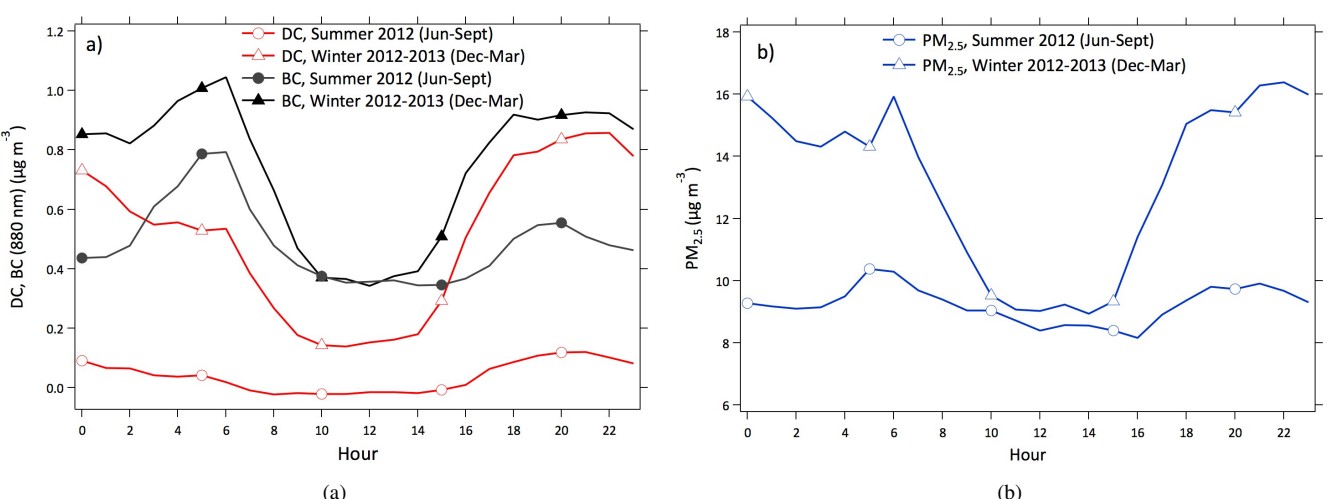

**Figure 4.** Diurnal plots (i.e., averaged into 24 hours) of a) DC (i.e., BC (370 nm)-BC (880 nm)) and BC (880 nm), and b) PM$_{2.5}$ values measured at the Rutland site from October 2011 to June 2013.

### 3.3   DC as a semi-quantitative marker for woodsmoke PM

Under woodsmoke-dominated environments we were studying, woodsmoke is the leading source of PM$_{2.5}$. Thus, we explored

in this section the relationships between measured PM$_{2.5}$ and DC to assess whether DC can be used as semi-quantitative predictor of woodsmoke PM$_{2.5}$, for both ambient and plume environments. We used the terms "semi-quantitative" for two reasons. One is that both highly time-resolved PM$_{2.5}$ and BC measurements contains significant uncertainties. The other reason is that the DC cannot be quantitatively interpreted as an exact amount of a specific compound unless the mixture of UV-absorbing species remains constant enough and an average absorption cross-section can be assumed.





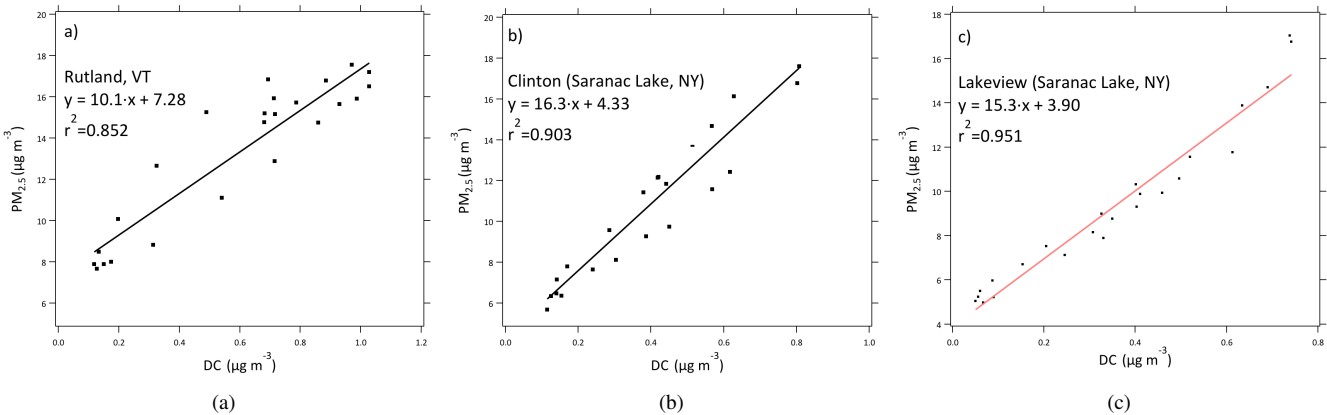

**Figure 5.** Diurnal $PM_{2.5}$ vs DC (i.e., BC (370 nm)-BC (880 nm)) averaged over the wintertime operation periods for a) the Rutland site, and over the entire operation periods for b) the Clinton site and c) the Lakeview site, respectively, into 24 hours.

### 3.3.1 Ambient environments (Rutland, Clinton and Lakeview)

Figure 5 depicts $PM_{2.5}$ vs DC for the three ambient sites, where we averaged all the hourly data, binned by hours of the day (i.e., 24 data points), over the wintertime operation periods for Rutland and over the entire operation periods for Clinton and Lakeview, respectively. The slopes derived from the linear regressions represent $\Delta$(Ambient $PM_{2.5}$)/$\Delta$DC. Table 4 presents

5  the linear regression results with all correlation coefficients of determination exceeding 0.85, which indicates strong positive correlations between changes in DC and changes in ambient $PM_{2.5}$ changes at the three sites. The most plausible explanation is that DC represents the woodsmoke PM, which typically have a strongly diurnal pattern, considering that wood burning and traffic are the only two major PM emissions sources, and wood burning is the only source of DC.

Furthermore, the regression coefficients for Clinton and Lakeview, the two ambient sites in Saranac Lake, NY, were very

10  similar, suggesting that the $\Delta$(Ambient $PM_{2.5}$)/$\Delta$DC is reproducible for similar ambient environments. However, the same relationship did not hold for the different environment of Rutland. The inclusion of two heating seasons for the Rutland site, compared to one season in Clinton and Lakeview, may have also contributed to the discrepancy.

**Table 4.** Semi-quantitative relationship between Delta-C ($\mu$g m$^{-3}$) and $PM_{2.5}$ ($\mu$g m$^{-3}$) in woodsmoke dominated ambient environments. The values inside the parentheses represent the corresponding one standard deviation.

| Site | | Regression | $r^2$ |
|---|---|---|---|
| Rutland, VT | | $PM_{2.5} = 10.1(\pm0.90)\cdot DC + 7.28(\pm0.60)$ | 0.852 |
| Saranac Lake, NY | Clinton | $PM_{2.5} = 16.3(\pm1.14)\cdot DC + 4.33(\pm0.52)$ | 0.903 |
| | Lakeview | $PM_{2.5} = 15.3(\pm0.74)\cdot DC + 3.85(\pm0.31)$ | 0.951 |





### 3.3.2 Plume environments (Ithaca)

Figure 6 presents the $PM_{2.5}$-DC relationships from two reoccurring woodsmoke sources based on the plume measurements, reported as 5-second moving averages, that were conducted in Ithaca, NY. Figure 6a-6d, and Figure 6e-6f characterized two sources different days, respectively. Both sources were woodstoves as the configurations of the exterior stacks were consistent
with this type of heating equipment. We estimated the background $PM_{2.5}$ concentrations for each day, and the values were $\sim$ 3 $\mu$g m$^{-3}$. Thus, we only included data points with $PM_{2.5}$ concentrations larger than 5 $\mu$g m$^{-3}$ in Figure 6 in order to capture the plume signals. The slopes derived from the linear regressions represent $\Delta$(Woodsmoke $PM_{2.5}$)/$\Delta$DC, as we conducted sampling in woodsmoke plume environments.

Overall, we observed a dominant set of correlated measurements, likely representing the average woodstove combustion con-
ditions, on each day. On both Figure 6c and 6f, "Condition 2" marked data points that define a different correlation are plotted with different symbols and a separate regression line. Each "Condition 2" line consisted of plume data recorded continuously. Possibly, during those conditions the woodstove combustion had been disturbed for some reasons (such as reloading the stove) for both Sources 1 and 2, thus significant deviation from the average conditions (denoted as "Condition 1" on both Figure 6c and 6f). For both Conditions 1 and Conditions 2, the correlations are generally strong. PM vs. DC slopes vary significantly
for individual sources (from 3 to 9.6 for Source 1, and from 7.4 to 28.6 for Source 2). Even for the same source, the slopes can change considerably during different operating conditions. Our analysis also suggests that the $PM_{2.5}$-DC relationships can be potentially utilized to distinguish different combustion and operating conditions of woodsmoke sources. It is expected that cleaner burns would have a larger slope – less organic aerosol per unit woodsmoke PM (Chandrasekaran et al., 2011, 2013). However, further studies are needed to link the $PM_{2.5}$-DC relationships to specific conditions.

### 3.3.3 DC and Heating degree days

Heating degree days (HDD), counted as the number of degrees that the daily average ambient temperature (F) is below 65°F, have been shown to be a better way to estimate energy use for space heating than actual temperature, as most homes or facilities are maintained at a temperature above 65°F. In a woodsmoke-dominated environment, we expected more woodsmoke with higher HDD.

We calculated the monthly average HDD for Rutland using the temperature data recorded at the weather station located in the Rutland-Southern Vermont Regional Airport (KRUT). In our analysis, DC/BC was adopted as a semi-quantitative woodsmoke emission indicator.

The rationale to use DC/BC, rather than DC directly, was to take BC as a dilution indicator to normalize DC. Even though the absolute values of DC change with meteorological conditions, DC/BC should be driven by the amount of woodsmoke PM
emissions generated, not woodsmoke PM concentrations.

Figure 7 illustrates the relationship between DC/BC and HDD, both presented as monthly averaged values. We observed a relatively strong linear relationship between DC/BC, which is an indicator for woodsmoke PM emissions, and HDD, which is a surrogate for space heating energy use. In other words, Figure 7 reveals not only a qualitative relationship (i.e., colder





(a)

(b)

(c)

(d)

(e)

(f)

**Figure 6.** PM2.5 vs. DC relationships from two reoccurring woodsmoke sources based on the plume measurements conducted in Ithaca, NY. Data are reported as 5-second averages. The dates are expressed in YYYY/MM/DD. The values inside the parentheses represent the corresponding one standard deviation.





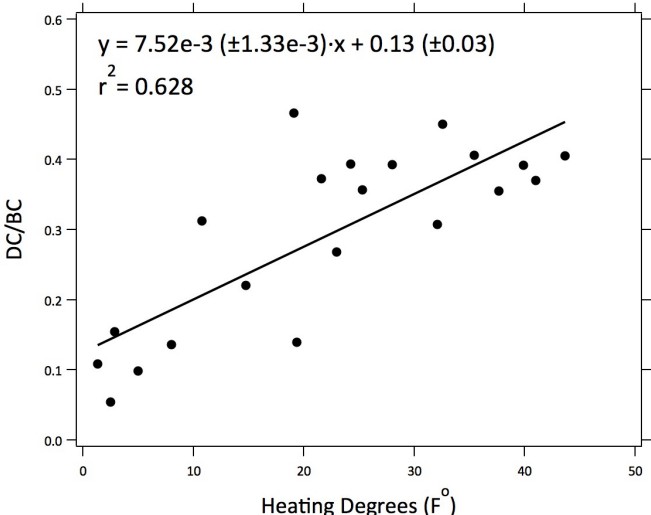

**Figure 7.** The relationship between DC/BC and HDD, both presented as monthly averaged values based on Rutland data. DC/BC is proposed as a woodsmoke PM emission indicator. The values inside the parentheses represent the corresponding one standard deviation.

the weather, the more woodsmoke PM), but also a potentially semi-quantitative relationship linking space heating energy and woodsmoke PM emissions. Note that the proportionality between DC/BC and HDD will vary from place to place, depending on various factors such as fraction of heating obtained from biomass, and types of biomass fuels burned.

### 3.3.4   Comparison between PAH and DC as a potential woodsmoke marker

Woodsmoke is known to contain polycyclic aromatic hydrocarbons (PAH)(Weimer et al., 2008), and large molecular weight PAHs have been suggested to contribute to light absorption by organic carbon in wood combustion (Chen and Bond, 2010). The simultaneous measurements of DC, $PM_{2.5}$ and PAH allowed us to conduct a preliminary comparison between particle-bound PAH as measured by the Ecochem PAS2000 and DC as a potential woodsmoke marker.

Figure 8 presents the correlations between diurnal average $PM_{2.5}$ and PAH for the Clinton (Figure 8a) and Lakeview (Figure

8b), compared to the similar plots for $PM_{2.5}$ vs. DC at the same sites (Figure 5b for Clinton and Figure 5c for Lakeview, respectively). Using the same method applied to Figure 5, we averaged the hourly $PM_{2.5}$ and PAH data over the entire operation periods into 24 hours. Overall DC was highly correlated with $PM_{2.5}$ ($r^2$ 0.9 for Clinton and 0.95 for Lakeview, respectively), whereas the correlation was much lower between $PM_{2.5}$ and particle-bound PAH much closer than PAH ($r^2$ 0.57 for Clinton and 0.27 for Lakeview, respectively). The preliminary comparison suggests that DC is a better woodsmoke indicator than PAH.

However, this finding was inconclusive since PAHs contain a large number of individual components, and what we have shown here were those measured by the particular instrument we deployed (i.e., EcoCHEM PAS2000), which may not capture the large molecular weight PAHs.




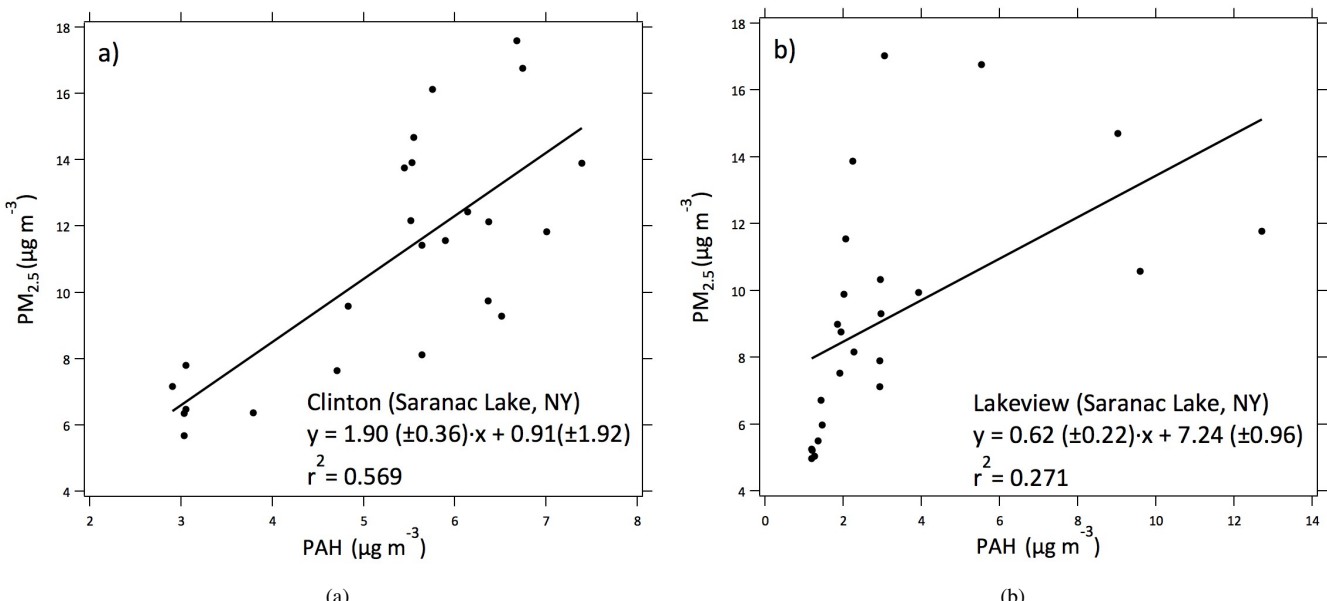

**Figure 8.** Diurnal average PM2.5 vs. PAH measured by EcoCHEM PAS2000 for a) the Clinton and b) Lakeview sites. We averaged the hourly $PM_{2.5}$ and PAH data over the entire operation periods at the two sites into 24 hours, respectively. The values inside the parentheses represent the corresponding one standard deviation.

## 4  Conclusions

We presented the results from the joint wintertime measurements of $PM_{2.5}$ and light-absorptive PM in woodsmoke-dominated ambient and plume environments in three Northeastern U.S. cities/towns, where other types of sources contributing to DC such as uncontrolled coal and kerosene burnings are usually rare. Our main conclusion is that DC can be a useful woodsmoke PM

marker, both qualitatively and semi-quantitatively. As a qualitative marker, DC can track the diurnal and seasonal woodsmoke PM patterns. As a semi-quantitative marker, DC can be used to estimate the amounts of woodsmoke PM. $PM_{2.5}$ vs DC relationship has been shown to be reproducible for similar ambient environments (like the Clinton and Lakeview sites in Saranac Lake, NY), but the same relationship did not hold for the different environment (like Rutland, VT). In other words, the relationship depends on the environment and combustion conditions.

This paper also presented several other potentially interesting findings: the $PM_{2.5}$-DC relationships can be utilized to distinguish different combustion and operating conditions of woodsmoke sources; the semi-quantitative relationship between DC vs. HDD could link space heating energy and woodsmoke PM emissions; DC tracks woodsmoke PM better than PAH measured by EcoCHEM PAS2000. Those findings could have important implications and applications in air quality management. However, as elaborated in the paper, further studies are needed to elucidate those findings.

*Competing interests.*  There are no competing interests.



*Acknowledgements.* The authors acknowledge funding support from the New York State Energy Research and Development Authority (NYSERDA) contracts #32974, 63035 and 63036, and appreciate the assistance of Aleshka Carrion-Matta, Neng Ji and Ye Lin Kim at Cornell University with conducting the field measurements. The New York State Department of Environmental Conservation provided forecasting support for mobile measurements, and the authors thank Robert Gaza, John Kent and Julia Stuart for their kind assistance. The

5    authors also thank Magee Scientific for loaning the Aethalometer Model AE-33 employed in the field measurements.



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
