# Peer review of "Joint Measurements of PM2.5 and light-absorptive PM in woodsmoke-dominated ambient and plume environments"

_Atmospheric Chemistry and Physics, 2017_

## Referee Comment (RC1) · Anonymous Referee #2 · 17 May 2017

This paper presents analysis that advances understanding of how to potentially assess woodsmoke in air without the need to conduct complex chemical analysis of a woodsmoke marker - levoglucosan. It presents mostly useful and straightforward analysis from a study using aethalometers to measure PM2.5 optical absorption under conditions in which mainly woodsmoke is expected. They use this data, along with other measurements to qualitatively and semi-quantitatively assess woodsmoke contribution to PM2.5 levels.

I found the analyses focused on the three fixed site data collection straightforward and the conclusions well-supported. I think Figure 7, DC/BC vs Heating Days is particularly compelling. Why is it only shown for Rutland site? It would be interesting to understand

how this stable this relationship is.

How was the CO2 data used?

There are two parts of this paper that are weak and I would recommend removal. The mobile monitoring in Ithaca is not well integrated and it does not add significant value to the paper. Likewise the PAH comparison is very lightly discussed and basically dismissed by the authors themselves. I recommend that these sections be removed so that the main point of the paper, the DC/BC analysis, is clear.
* * *

---

## Referee Comment (RC2) · Anonymous Referee #1 · 29 May 2017

The authors present a compilation of interesting field campaigns in two winters in the NE US in environments where wood burning is an important source of air pollution. PM2.5 and Black Carbon (BC) have been measured and the delta-C (DC) parameter identified as a good (semi)quantitative indicator for the presence of wood burning contribution to PM2.5.

The manuscript is an analysis of local particular air pollution. The following changes and additions will greatly contribute to its scientific significance and quality and will facilitate the understanding by an average reader.

The authors are interested in quantification of the contribution of woodsmoke to PM2.5. Source apportionment always depends on the assumptions of the method employed

to determine the sources. The authors make the statement that PM2.5 is dominated by woodsmoke on p. 5 (lines 12-13) and p. 10 (line 14). This should be the result of the work, not an initial assumption, which cannot be tested without additional chemical methods being applied to the same samples. The starting assumptions need to be checked and a "proof" that PM2.5 is exclusively due to woodburning needs to be produced. Alternatively, a reference to a monitoring agency report or previous publications would be advantageous.

BC and PM2.5 have similar diurnal variation. We can see from Figures 3 and 4 that the BC/PM ratio is not constant during the day. This means that the composition of PM is changing. The authors correctly point out that unless UV-absorbing species feature a constant absorption cross section, DC cannot be considered quantitative. The change in composition implies the change in the absorption cross section. This part of section 3.3 needs to be expanded and arguments provided. Comparison to other source apportionment methods using similar methods would help (Sandradewi et al., 2008).

Finally, the manuscript ignores the production of SOA. The change in composition of PM is already evident from the diurnal variation of the BC/PM ratio. SOA and the applicability of the method described in the manuscript need to be discussed in terms of primary and secondary PM. SOA can dominate PM in places where PM is heavily impacted by wood burning – the sites presented in the articles are such places.

I agree with Reviewer 2 that the presentation of the "PAH" as measured in this campaign is weak. I would recommend to either remove this subsection or significantly redact it and expand the discussion on the PAH measurement method. If the authors expand the section, they should switch the PM2.5 and PAH axes. They argue that PAH is not as good an indicator as DC, hence the plot should be made into PAH as a constituent of PM2.5 by switching the axes.

Minor comments:

Dates need to be changed so that the European readers will find them unambiguous (Figure 3).

Figure 7: Why just report this for Rutland?

References

Sandradewi, J., Prevot, A. S. H., Szidat, S., Perron, N., Alfarra, M. R., Lanz, V. A., Weingartner, E., and Baltensperger, U.: Using aerosol light absorption measurements for the quantitative determination of wood burning and traffic emission contributions to particulate matter, Environ. Sci. Technol., 42, 3316-3323, doi: 10.1021/es702253m, 2008.

---

## Author Comment (AC1) · 16 Jun 2017

We greatly appreciate the valuable comments from Anonymous Referee #1. In addition to the replies below, one updated figure, revised manuscript with marked changes are also enclosed as a supplement.

1. "The authors are interested in quantification of the contribution of woodsmoke to PM2.5. Source apportionment always depends on the assumptions of the method employed to determine the sources. The authors make the statement that PM2.5 is dominated by woodsmoke on p. 5 (lines 12-13) and p. 10 (line 14). This should be the result of the work, not an initial assumption, which cannot be tested without additional chemical methods being applied to the same samples. The starting assumptions need to be checked and a "proof" that PM2.5 is exclusively due to woodburning needs to be produced. Alternatively, a reference to a monitoring agency report or previous publications would be advantageous. "

The authors would like to acknowledge the vague definition of "woodsmoke-dominated". In the original manuscript, we cited a report from the State of Vermont showing the wood is the dominant heating fuel in Rutland (Frederick and Jaramillo, 2016) and the 2014 National Emission Inventory (NEI) to show that woodsmoke is a dominating PM emission source. We have provided more quantitative information in the revised manuscript to make the point. The NEI only provide county-level annual emission inventories. The town of Saranac Lake spans two counties in New York, i.e., Essex and Franklin. The point we try to make is that woodsmoke emission is the pre-dominant source of wintertime PM2.5 emissions in both Rutland and Saranac Lake. For the plume environments in Ithaca, NY, it is woodsmoke-dominant in nature as we purposefully sampled woodsmoke plumes.

In Section 2.1 of the revised manuscript, we added, "According to the 2014 National Emission Inventory, residential woood combustion (RWC) contributes to approximately 38.6% of the annual PM2.5 emissions in Rutland County. In comparison, on-road mobile sources only account for 1.4%. Considering the seasonal patterns of various emission sources, it is clear that RWC is the predominant primary PM2.5 source in Rutland during wintertime." and "The 2014 National Emission Inventory indicated that RWC accounts for approximately 22.4 to 25.4% of the annual PM2.5 emissions, while the contribution of on-road mobile sources is between 2.8 to 3.9%, which indicated that it is also a woodsmoke-dominated environment during wintertime."

In addition, the main goal of our study is not a source apportionment in the three reported towns/cities. As described above, even on annual basis, woodsmoke PM2.5 emissions is 10 to 40 times higher than mobile emissions in Rutland and Saranac Lake. During wintertime, we expect the woodsmoke emission would exceed mobile sources

by over 100 times. Our primary goal is to address the concerns whether DC (aka Delta-C) is a useful woodsmoke marker for air quality management, in both qualitative and semi-quantitative sense. As presented in the last paragraph of the introduction section, "Our study can be regarded as a "necessary condition test" for DC serving as a woodsmoke PM marker. In other words, DC would be deemed an inappropriate marker if it were unable to track woodsmoke PM patterns even under woodsmoke-dominated environments."

2. "BC and PM2.5 have similar diurnal variation. We can see from Figures 3 and 4 that the BC/PM ratio is not constant during the day. This means that the composition of PM is changing. The authors correctly point out that unless UV-absorbing species feature a constant absorption cross section, DC cannot be considered quantitative. The change in composition implies the change in the absorption cross section. This part of section 3.3 needs to be expanded and arguments provided. Comparison to other source apportionment methods using similar methods would help (Sandradewi et al., 2008)."

We agree with the reviewer that the varying BC/PM ratio may indicate changing composition as well as changing absorption cross section. The plume data presented in Figure 6 also imply varying absorption cross section with combustion conditions. The main message in Figure 5 is that averaging stationary PM and BC data over a long period of time (e.g., over a winter month or longer in a fixed location) may lead to an average absorption cross section, i.e., a constant $\Delta$(Ambient PM2.5)/$\Delta$DC. We have revised the first paragraph in Section 3.3.1 by adding the following sentences, "Furthermore, Figure 5 suggests that averaging stationary PM and BC data over a long period of time (e.g., over a winter month or longer in a fixed location) may lead to an average absorption cross section, i.e., a constant $\Delta$(Ambient PM2.5)/$\Delta$DC, even though PM composition and the resulting absorption cross section may vary with time."

The research team initially planned to apply the method presented by Sandradewi et al. (2008ab) to the ambient data. However, as described in Section 1 of the original manuscript, that method often requires light absorption measurements at multiple wavelengths to have a reliable estimate on $\alpha$. Since the ambient data to be presented in this paper were collected by a two-wavelength Aethalometer, we did not attempt to calculate $\alpha$ from the ambient data. Note that we employed a seven-wavelength Aethalometer in the plume measurement, which allowed us to apply the method of Sandradewi et al. (2008). We will report the related findings in a separate publication. We further revised the related paragraph in the Introduction to elaborate our rationale: "Another approach taking advantage of UV enhancement (or wavelength dependence of the aerosol absorption coefficient in general), as reported by Sandradewi et al. (2008a), derives light absorption Ångström exponents ($\alpha$) from multi-wavelength Aethalometer readings. $\alpha$ is close to 1 for traffic sources, and varies for woodsmoke, but generally much larger than 1. Assuming certain value of $\alpha$ for woodsmoke, Sandradewi et al. (2008b) conducted quantitative analysis of source contributions to PM. This approach often requires light absorption measurements at multiple wavelengths to have a reliable estimate on $\alpha$ (Chen et al., 2015). Sandradewi et al. (2008b) showed that using different pairs of wavelengths led to different values of $\alpha$ for woodsmoke. Since the ambient data to be presented in this paper were collected by a two-wavelength Aethalometer, we did not attempt to calculate $\alpha$. Given the uncertainties associated with values of $\alpha$ for woodsmoke for our study, we did not perform the source apportionment analysis similar to that presented by (Sandradewi et al., 2008a)."

Furthermore, as described in our reply to Comment 1, we purposefully selected woodsmoke-dominated environments to conduct a necessary condition test for DC as a woodsmoke marker. Our study does not directly address whether DC is a good woodsmoke maker in environments not dominated by woodsmoke, where source apportionment is probably necessary.

3. "Finally, the manuscript ignores the production of SOA. The change in composition of PM is already evident from the diurnal variation of the BC/PM ratio. SOA and the applicability of the method described in the manuscript need to be discussed in terms

of primary and secondary PM. SOA can dominate PM in places where PM is heavily impacted by wood burning – the sites presented in the articles are such places."

In the original manuscript, we described our thinking why SOA does not look like a main driver for DC. We revised the related discussions to make our point more clear. The revised discussions appear near the end of the second paragraph in Section 3.2: "As mentioned earlier, previous studies found that SOA products may result in DC signals. If SOA formation were significant, we would expect that PM2.5 and/or DC would peak around mid-day. The distinct diurnal patterns illustrated in Figure 4 is more consistent with strong influence of local emissions. Moreover, the seasonal trend shown in Figure 3 indicates that DC peaked during wintertime when SOA production is small and approached zero during summertime when SOA production is expected to be high. Therefore, both the diurnal and seasonal patterns indicate that SOA is not a main driver for DC in Rutland."

4. "I agree with Reviewer 2 that the presentation of the "PAH" as measured in this campaign is weak. I would recommend to either remove this subsection or significantly redact it and expand the discussion on the PAH measurement method. If the authors expand the section, they should switch the PM2.5 and PAH axes. They argue that PAH is not as good an indicator as DC, hence the plot should be made into PAH as a constituent of PM2.5 by switching the axes."

We have selected the referred section in the revised manuscript.

5. "Dates need to be changed so that the European readers will find them unambiguous (Figure 3)."

The revised figure is enclosed and included in the revised manuscript.

6. "Figure 7: Why just report this for Rutland?"

Among the sites included in our study, only Rutland had DC data over a year. We will search for more data in other locations, and likely report the findings in a separate

publication. For long-term continuous light-absorption measurement in a woodsmoke-dominated environment, Rutland is probably one of very few in the U.S.

Please also note the supplement to this comment:
http://www.atmos-chem-phys-discuss.net/acp-2017-213/acp-2017-213-AC1-supplement.pdf

———————————————————

[Figure]

[Figure]

**Fig. 1.**

---

## Author Comment (AC2) · 16 Jun 2017

We greatly appreciate the valuable comments from Anonymous Referee #2. In addition to the replies below, revised manuscript with marked changes are also enclosed as a supplement.

1. "I found the analyses focused on the three fixed site data collection straightforward and the conclusions well-supported. I think Figure 7, DC/BC vs Heating Days is particularly compelling. Why is it only shown for Rutland site? It would be interesting to understand how this stable this relationship is."

[Figure]

Among the sites included in our study, only Rutland had DC data over a year. We will search for more data in other locations, and likely report the findings in a separate publication. For long-term continuous light-absorption measurement in a woodsmoke-dominated environment, Rutland is probably one of very few in the U.S.

2. "How was the CO2 data used?"

We did not use the CO2 data in the current manuscript. The data did not pass our quality assurance (QA) check. We are still trying to figure out the possible causes.

3. "There are two parts of this paper that are weak and I would recommend removal. The mobile monitoring in Ithaca is not well integrated and it does not add significant value to the paper. Likewise the PAH comparison is very lightly discussed and basically dismissed by the authors themselves. I recommend that these sections be removed so that the main point of the paper, the DC/BC analysis, is clear. "

We have removed the section discussing PAH measurement. We'd like to explain why we think the Ithaca plume analysis is inherent part of the paper. The main goal of our manuscript is to address the concerns whether DC (aka Delta-C) is a useful woodsmoke marker for air quality management, in both qualitative and semi-quantitative sense. By studying woodsmoke plume data, we showed linear relationships between PM2.5 and DC can be used to distinguish different combustion conditions. The fundemental principle is that different combustion conditions lead to different PM composition, which in turns lead to different absorption cross sections. The high time resolution AE-33 we deployed can capture the changes in absorption cross sections, which can potentially allows us to track combustion conditions. We added the following sentence near the end of the second paragraph of Section 3.3.2, "In other words, the different combustion conditions lead to different chemical compositions and absorption cross sections, which can be potentially captured by high time resolution light absorption measurements."

Please also note the supplement to this comment:
http://www.atmos-chem-phys-discuss.net/acp-2017-213/acp-2017-213-AC2-supplement.pdf
* * *
[Figure]

**Supplement:**

[revised manuscript text omitted]

---

## Author Comment (AC3) · 16 Jun 2017

The reply should be:

We have REMOVED the referred section in the revised manuscript.

---

## Author Response (AR2)

**Replies to Comments from the Co-Editor**

We greatly appreciate the valuable comments from the Co-Editor. Our replies are attached below.

1. *However, one important issue remains: the DC is closely associated with sources (this is the argument you use in the introduction) and an apportionment needs to be shown and compared to. The argument that you use while claiming that the Sandradewi et al (2008) source apportionment cannot be used (that AAE_wood burning too variable) is weak. The method clearly depends on the assumed (!) AAE_wb and AAE_ff values and a sensitivity analysis need to be performed. Zotter et al. (2017) have shown both the influence of the choice of AAE and the sensitivities and have validated the values using C14. Please add the Sandradewi et al. source apportionment for comparison and perform a sensitivity analysis.*

We acknowledged the statement that *the Sandradewi et al (2008) source apportionment cannot be used is too strong.* In the revised manuscript, we calculated the AAE values based on the averaged diurnal profiles of BC(370 nm) and BC(880nm) data for the summer and winter months in Rutland, VT. We also marked the recommended AAE values for traffic and woodsmoke by Zotter et al. (2017). The figure below illustrates the results, which suggests qualitatively that winter PM, especially at night, is dominated by woodsmoke, while summer PM is dominated by traffic.

[Figure]

The AAE figure also indicates the mismatch between calculated AAE and recommended AAE value for woodsmoke (1.68). The calculated AAE values after 6 pm during wintertime are actually greater than 1.68. If 1.68 were used as AAE for woodsmoke in a source apportionment analysis, more than 100% of the PM would come from wood burning. Of course, setting woodsmoke AAE value higher (e.g., 2) would give different source apportionment results. But the fact that winter AAE values are much greater than summer AAE values in Rutland is strongly implies the seasonal differences in dominant PM sources, consistent with the emission inventory results we cited in the manuscript. Furthermore, we would have to choose parameters relating the light absorption to the PM mass (i.e., "c1" and "c2" in Sandradewi et al. (2008)), which will introduce another layer of uncertainties.

Therefore, we revised Section 3.2 in the original manuscript by adding the following paragraph:

> Figure 5 depicts the diurnal profiles of AAE (also known as α), derived from the two-wavelength AE-21 (i.e., 370 nm and 880 nm) data in Rutland, for both summer months (July to September 2012) and winter months (December 2012 to March 2013). Overall, the values of α in the winter months (ranging from 1.37 to 1.76) are much greater than those in the summer months (ranging from 0.93 to 1.24). Zotter et al. (2017) recommended α values for traffic and woodsmoke as 0.9 and 1.68, respectively, by comparing the source apportionment of equivalent black carbon using the Aethalometer model originally proposed by Sandradewi et al. (2008a, b) with $^{14}$C measurements of the elemental carbon fraction from several locations and campaigns across Switzerland. Those α values are also marked in Figure 5. Therefore, Figure 5 suggests, qualitatively, that woodsmoke PM dominates during the winter months, while traffic (or fossil fuel combustion) PM is a major source of PM during the summer months, which is consistent with the findings based on the emission inventory described earlier.

2. *Please take the comments of reviewer 1 into account in the updated manuscript. You find them attached below.*

We have prepared replies to the comments by Reviewer 1 starting from the next page.

[revised manuscript text omitted]